# MCAM/MUC18/CD146 as a Multifaceted Warning Marker of Melanoma Progression in Liquid Biopsy

**DOI:** 10.3390/ijms222212416

**Published:** 2021-11-17

**Authors:** Maria Cristina Rapanotti, Elisa Cugini, Marzia Nuccetelli, Alessandro Terrinoni, Cosimo Di Raimondo, Paolo Lombardo, Gaetana Costanza, Terenzio Cosio, Piero Rossi, Augusto Orlandi, Elena Campione, Sergio Bernardini, Marcel Blot-Chabaud, Luca Bianchi

**Affiliations:** 1Department of Onco-Haematology, University of Rome Tor Vergata, Viale Oxford 81, 00133 Rome, Italy; 2Department of Laboratory Medicine, University of Rome Tor Vergata, Viale Oxford 81, 00133 Rome, Italy; elisa.cgn@gmail.com (E.C.); marzianuccetelli@yahoo.com (M.N.); alessandro.terrinoni@uniroma2.it (A.T.); costanza@med.uniroma2.it (G.C.); bernardini@med.uniroma2.it (S.B.); 3Dermatology Unit, Department of Systems Medicine, University of Rome Tor Vergata, Via Montpellier 1, 00133 Rome, Italy; cosimodiraimondo@gmail.com (C.D.R.); lombardo.paolo89@gmail.com (P.L.); terenziocosio@gmail.com (T.C.); elena.campione@uniroma2.it (E.C.); luca.bianchi@uniroma2.it (L.B.); 4Department of Surgery Sciences, University of Rome Tor Vergata, Via Montpellier 1, 00133 Rome, Italy; piero.rossi00133@uniroma2.it; 5Anatomic Pathology, University of Rome Tor Vergata, Via Montpellier 1, 00133 Rome, Italy; orlandi@uniroma2.it; 6Institut National de la Sante et de la Recherche Medicale (INSERM), UMR-S 1076, Aix-Marseille University, UFR Pharmacy, 13005 Marseille, France; marcel.blot-chabaud@laposte.net

**Keywords:** MCAM/MUC18/CD146, soluble CD146, liquid biopsy, circulating melanoma cells (CMCs), gene-expression panel

## Abstract

Human malignant melanoma shows a high rate of mortality after metastasization, and its incidence is continuously rising worldwide. Several studies have suggested that MCAM/MUC18/CD146 plays an important role in the progression of this malignant disease. MCAM/MUC18/CD146 is a typical single-spanning transmembrane glycoprotein, existing as two membrane isoforms, long and short, and an additional soluble form, sCD146. We previously documented that molecular MCAM/MUC18/CD146 expression is strongly associated with disease progression. Recently, we showed that MCAM/MUC18/CD146 and ABCB5 can serve as melanoma-specific-targets in the selection of highly primitive circulating melanoma cells, and constitute putative proteins associated with disease spreading progression. Here, we analyzed CD146 molecular expression at onset or at disease recurrence in an enlarged melanoma case series. For some patients, we also performed the time courses of molecular monitoring. Moreover, we explored the role of soluble CD146 in different cohorts of melanoma patients at onset or disease progression, rather than in clinical remission, undergoing immune therapy or free from any clinical treatment. We showed that MCAM/MUC18/CD146 can be considered as: (1) a membrane antigen suitable for identification and enrichment in melanoma liquid biopsy; (2) a highly effective molecular “warning” marker for minimal residual disease monitoring; and (3) a soluble protein index of inflammation and putative response to therapeutic treatments.

## 1. Introduction

Identification and characterization of cell-adhesion molecules that play a role during melanoma progression may represent a specific target for diagnosis and clinical therapy.

Among this class of molecules with a role in melanoma, and radial growth phase (RGP) (horizontal expansion) to vertical growth phase (VGP) evolution, MCAM/MUC18/CD146, also cited as A32 antigen, MelCAM, or S-Endo-1, has been recently identified [1,2,3], and we will refer to this protein as CD146 throughout this entire manuscript.

This protein is a 113 kDa transmembrane glycoprotein that belongs to the immunoglobulin superfamily, and it is mainly expressed at the intercellular junction of endothelial cells, where it interacts with VEGFR-2 [1,2,3]. It was firstly identified in melanoma cells as a cell-adhesion molecule. It is known to be a cell-surface receptor of various ligands involved in development, signal transduction, cell migration, mesenchymal stem cell differentiation, angiogenesis, and immune response [1,2,3,4,5,6]. The human form is composed of an extracellular portion, containing five Ig-like domains with two variable regions (V) and three constant regions (C2) (V-V-C2-C2-C2); a single transmembrane domain; and a cytoplasmic domain containing a single tyrosine residue. CD146 glycosylation, in malignant melanoma, is mainly represented by β-1,3-galactosyl-glycosil-glycoprotein and β-1,6-*N*-acetylglucosaminyltransferase-3. The degree of CD146 glycosylation appears to be directly related to malignant progression of tumors [6,7,8,9,10,11,12,13,14,15,16]. Alternative splicing of the transcript, with the junction of exon 14 to exon 16, generates two membrane-isoforms of CD146, lgCD146 (long) and shCD146 (short) CD146. The two proteins differ by the C-terminal (cytoplasmic) region, since the splicing modifies the open-reading frame of exon 16. The lgCD146 displays two phosphorylation sites for protein kinase (PKC) and double leucine motifs for baso-lateral membrane targeting, while the shCD146 isoform contains one PKC and one PDZ-binding domain, suggesting an involvement in cell signaling [17,18]. In endothelial cells, lgCD146 is distributed at the cell junction and is involved in structural functions, while shCD146 is expressed in the cell apical pole, contributing to the angiogenetic process [16,17,18,19,20,21,22,23].

An additional soluble form, sCD146, was described firstly by Bardin et al., 1998 [20]. It is constituted only by the extracellular portion of the molecule, and is generated by a proteolytic cleavage through metalloproteinases (MMPs). It can be detected in biological fluids, including serum, urine, and cerebrospinal fluid. A schematic diagram of the CD146 protein is shown in Figure 1.

In normal adults, it is expressed in different tissues, and in particular in the different cells constituting the vessels; i.e., endothelial cells, smooth muscle cells, and pericytes. In pathology, CD146 correlates with tumor thickness and metastatic potential of human melanoma cells in mice and humans [7,8,9,10,11,12,13,14]. Overexpression studies have shown that it affects angiogenesis and promotes neoplastic progression from local invasive to metastatic. This occurs through an upregulation of metalloproteinase (MMP-2) and an interaction between the extracellular matrix and vascular endothelium. CD146 can be considered as a key oncogene in driving melanoma progression and metastasis, particularly with respect to vascular and lymphatic metastasis [13,14,15,16].

Elevated membrane CD146 expression and high soluble CD146 concentration in plasma (range of 200 to 400 ng/mL in healthy subjects) are often correlated and are associated with pathological conditions, such as inflammation and tumorigenesis [24,25]. Thus, increased CD146 overexpression has been reported in inflammatory lesions compared to the normal status as inflammatory bowel disease [26], spondylarthritis synovium [27], asthma [28], atherosclerotic plaque formation and progression [29], other autoimmune disorders [30,31,32,33,34], or systemic sclerosis [30,32].

Accumulating evidence has also shown that inflammation can promote melanoma progression through several mechanisms, such as cell proliferation, motility, metastatic dissemination, and angiogenesis [24,25].

Different antibodies targeting the membrane form of CD146 have been developed, such as the S-endo1 Ab, which reduces monocyte transmigration [35]. AA98 mAb reduced leukocyte transmigration in a multiple sclerosis mouse model [36,37], and also decreased tumor growth of human hepatocarcinoma and leiomyosarcoma cells, displaying an efficient inhibitory effect on tumor growth and angiogenesis [22,38]. The ABX-MA1 mAb reduced the growth and metastases of human melanoma cells in mouse models [38]. Unfortunately, targeting of membrane CD146 also leads to alteration of the vascular system, producing unwanted side effects on vascular integrity and physiological functions. Another monoclonal antibody against CD146, TsCD146 mAb [39], has been documented to specifically recognize CD146 expressed on cancer cells, but not CD146 expressed on physiological vessels, suggesting structural differences between “malign CD146” and “physiological CD146”. Noteworthy, soluble CD146 has been regarded as a poor prognostic factor in almost all solid tumors and autoimmune diseases. A monoclonal neutralizing antibody, M2J1, specifically targeting this soluble form has been recently developed. This antibody specifically reduced the development of CD146-positive melanoma in different mouse models by reducing cancer cells’ proliferation and metastasis and abolishing tumor-induced vascularization.

Thus, the demonstrated involvement of CD146 and sCD146 in melanoma progression and dissemination has led to the development of immune therapies against CD146 and sCD146 that could represent valuable strategies for fighting melanoma [40,41].

## 2. CD146 Molecular Expression as a Melanoma-Associated Marker in Peripheral Blood

Identification of specific tissue markers from the peripheral venous blood of patients would be of particular interest to detect and follow disease progression in melanoma. Theoretically, specific detection of “melanocytic transcripts” should correlate with metastatic spreading of cells from a primary tumor [42,43,44]. These cells are detectable in peripheral blood either soon after the surgical resection of primary tumors regardless of their thickness; in late-stage disease; and even in clinically disease-free patients [45].

We previously reported [45] that expression of CD146 in patients’ blood samples was associated with advanced stages of melanoma. A highly specific and sensitive multimarker RT-PCR assay based on the amplification (presence) of Tyr-OH, MART-1, MAGE-3, MUC-18/MCAM/CD146, and p97 transcripts was elaborated for analyzing 100 melanoma patients (AJCC stages I–IV). These melanoma-associated markers were selected for their well-known differentiation properties and expression frequency in melanoma, and for their high specificity in the RT-PCR assay [46,47,48,49,50,51,52,53,54,55,56,57,58]. The statistical evaluation of the molecular association of CD146 expression with advanced melanoma stages showed a significant correlation. The expression of CD146 was shown to increase the probability of evolution toward disease progression, as AJCC III–IV stages (American Joint Committee on Cancer) and higher recurrence incidence [59].

All these findings strongly support a reliable interest in CD146 in the detection of melanoma progression. Thus, we decided to extend our analysis to a larger series of patients (*n* = 175) exploring circulating CD146 expression by RT-PCR assay on serial blood samples obtained during the clinical course of the disease. Our investigation [58,59] emphasized a correlation between *CD146* mRNA detection (from whole peripheral blood) and degree of expression of this marker on the corresponding primary melanoma tissue, tumor thickness, AJCC stages, and clinical outcome. We showed that *CD146* mRNA expression correlated with melanoma diagnosis and progression of the disease. Either when detectable from the beginning of the pathology or subsequently acquired during the course of the disease, CD146 was significantly associated with a poor prognosis and death. The loss of CD146 mRNA expression during follow-up was associated with a clinically disease-free status. Comparison of the clinical outcome between early-AJCC-stage patients sharing fleeting expression and patients who later acquired a persisting expression showed a statistically significant difference. When considering patients at advanced stages, we observed a statistically significant difference between the poor clinical evolution and outcome of CD146-positive patients as compared to patients who never expressed this biomarker, who presented a good outcome or stable disease. Hence the presence of CD146 expression could be used as predictor of clinical relapse, and its absence associated with a stable disease/disease-free status. Thus, CD146 could represent a “molecular warning” of disease progression [58,59]. All these findings strongly supported the need for a reliable CMC detection in order to investigate CD146 as a specific viable melanoma marker in early and widespread metastasis.

## 3. CD146 as an Enrichment and Capture Marker for Circulating Melanoma Cells

To date, the liquid biopsy is renowned as important technology in the detection of cancer-related biomarkers, mostly for early diagnosis, screening, prognosis, analysis of minimal residual disease (MRD), and subsequent chance to design personalized therapy and patient monitoring. The liquid biopsy constitutes a noninvasive test to detect circulating tumor cells (CTCs) or the products of tumors, including proteins, cell-free DNA, and exosomes, and can be performed in various biological fluids (peripheral blood, urine, and ascites).

CTCs, with respect to other materials, have some advantages in clinical applications. They can be identified morphologically, and their genetic and molecular characterization can be performed by analyzing both potential tumor biomarkers and specifically selected DNA mutations.

In malignant melanoma (MM), CTCs are detectable in the peripheral blood soon after the surgical resection of the primary tumor, regardless of the thickness. They are also detectable in late stages or in clinically disease-free patients [60,61,62]. Measuring circulating melanoma cells (CMCs) before they become clinically detectable represents a potentially powerful method for monitoring patients with malignancies who present a minimal morbidity. Two studies showed that detection of two or more CMCs per 7.5 mL of blood was associated with shorter survival [63,64,65]. In carcinomas, immuno-magnetic enrichment is conventionally performed with epithelial surface markers such as EpCAM or cytokeratin antigens for a “positive” selection. The Food and Drug Administration has approved the CellSearch^®^ Circulating Tumor platform for the collection and isolation of CTCs of these carcinomas. At present, only EpCAM (Janssen Diagnostic, LLC, Raritan, NJ, USA) has been recognized. As already reported, CMCs’ lack a ubiquitous marker, since they do not express the common epithelial cell-surface marker EpCAM due to the origin of the melanocytes from the neural crest. Nevertheless, a variety of markers associated with some melanoma-specific cell-surface epitopes have been proposed, such as CD146 and MSCP/NG2, (melanoma-associated chondroitin sulfate) together with stem cell markers such as ABCB5 (ATP-binding cassette-subfamily member B) and CD271 [66,67]. The majority of cell-based liquid biopsy tests have been developed with magnetic-separation technologies by using magnetic nanoparticles (MNPs) that can bind specifically to the antigens on the target cells. Enhancement and specificity of capture efficiency is obtained by ablation of “nontumor” antigens and white blood cells (WBCs) through coating, reducing the interactions between the MNPs and the white blood cells and ranging in cell purity until 90–92%. Despite CMCs being phenotypically and molecularly heterogeneous, we decided to use CD146 as a capture antigen, as it is expressed up to 80% in MM, together with the melanoma-initiating marker ABCB5.

Preliminary findings documented the efficiency of enrichment and isolation of CMCs from plasma of a series of 21 patients affected by early or advanced stage melanoma, based on CD146 or ABCB5 expression, or a combination thereof [68].

A qualitative expression panel of melanoma aggression markers, contemplating the angiogenic-potential, melanoma-initiating, and melanoma-differentiation drivers, as well as cell–cell adhesion molecules and matrix metalloproteinases, was characterized at the mRNA level [68,69].

A significantly different expression of the s transcripts encoding for *CD146*, *MMPs,* and *VE-Cadh* was documented between and within the three CMC fractions enriched with CD146-, ABCB5- and both CD146/ABCB5-coated beads, when analyzing advanced-stage patients vs. early-stage patients. We could distinguish “endothelial” CMCs (CD45-CD146+ enriched; E-CMCs), “stem” CMCs (CD45-ABCB5+ enriched; S-CMCs), and hybrid biphenotypic “stem–mesenchymal” CMCs (CD45-CD146+/ABCB5+ enriched; SM-CMCs), as a function of three distinct gene-expression profiles. In particular, the E-CMCs were characterized by a positive expression of genes involved in migration and invasion, whilst the S-CMCs only expressed stem and differentiation markers. The third subpopulation, despite the contextual CD146 and ABCB5 enrichment, did not express *ABCB5,* and showed an invasive phenotype. Molecular expression of *CD146*, *MMPs,* and *VE-Cadh* was associated with disease progression and allowed the differentiation between high-risk versus low-risk patients. Main melanoma-associated markers as tyrosin-idroxilase (*Tyr-OH)*, *MelanA/MART1* antigen, or epithelial cell-adhesion molecules E-cadherin (*E-Cadh*) and N-cadherin *(N-Cadh)* were absent in all CMCs fractions. All three distinct CMCs subpopulations exhibited a primitive “stem–mesenchymal” profile, suggesting a highly aggressive and metastasizing phenotype.

An additional longitudinal monitoring study [70] performed on the three CMC subpopulations collected at baseline and during follow-up showed that persistency or acquisition of *CD146*, *VE-CADH,* and *MMPs* expression was associated with disease progression, highlighting association between these genes and disease-spreading progression and minimal residual disease. Conversely, a drastic expression shutdown of these proteins was documented when patients achieved clinical remission.

These studies confirmed the phenotypic and molecular heterogeneity observed in melanoma and highlighted three putative genes involved in early melanoma spreading and disease progression, decoding a “stem–mesenchymal-like phenotype” in all three subpopulations, and suggesting that this signature could be related to a metastatic behavior.

## 4. Current Findings: CD146 as an Enrichment and Capture Marker at Melanoma Onset or Disease Recurrence

As a function of these results, we decided to focus our studies only on the two cell populations, “endothelial” and “hybrid stem–mesenchymal”—subsets that represent the most promising tools for clinical investigation, as they express genes strongly associated with the potential ability to metastasize. So, we enlarged our population, collecting up to 30 “baseline-samples” that were split in two distinct subpopulations classified as E-CMCs and SM-CMCs. We assessed these samples with a smaller protein expression panel, excluding proteins that were poorly expressed in our cellular subsets, such as melanoma tissue differentiation markers *TyrOH* and *MelanA/MART1*, and the benign epithelial cell-adhesion molecule E-cadherin, *E-Cadh*. We also excluded the stem melanoma-stem marker ABCB5 that, despite the contextual effective CD146 and ABCB5 enrichment, was rare, even in the ABCB5-enriched CMC fraction [68,71].

So, all 60 cellular subsets were molecularly characterized by analyzing the following proangiogenic factors: vascular endothelial growth factor (*VEGF*); basic fibroblast growth factors (*bFGF*); cell–cell adhesion factors such as neuronal cadherin, (*N-Cadh CDH2*) and vascular endothelial cadherin (*VE-Cadh CDH5)*; endothelial antigen *CD146* isoforms (long, short, and 5′-portion); and matrix metalloproteinases (*MMP-2, MMP-9)* [68,69]. In this enlarged study, we also enrolled patients if staged AJCC ≥ pT1b with a confirmed diagnosis of histological and immune-histochemical malignant melanoma.

We then stratified all these samples collected at first observation into two disease categories: “clinically remission” patients (12) and “clinically evident disease” patients (18). Patients’ demographic (pts) and clinical characteristics are shown in Table 1.

Overall, 27 patients out of 30 (sensitivity of 90%), or rather 54 cellular subsets out of 60, were found positive for expression of one of the nine transcripts at least in one fraction from their blood (E-CMCs or SM-CMCs). The nine transcripts considering the two subpopulations, E-CMCs and SM-CMCs, were expressed as follows: *CD146* 5′-portion (45.7–30%), long isoform (56.6–53.3%), short isoform (46.0% both), *VEGF* (13.3–10.0%), *bFGF* (20% both), *VE-Cadh* (43.3% both), *N-Cadh* (33.3–26.6%), *MMP2* (56.6% both) and *MMP9* (50.0%) respectively. Table 2A shows the gene-expression frequencies of transcripts in the two distinct cellular subpopulations, E-CMCs and SM-CMCs, respectively, collected from 12 “clinically remission patients” and 18 “clinically evident disease” patients.

Overall, the “clinically evident disease” patients were characterized by higher gene expression compared to the “clinically remission” patients, as expected and already reported [68]. Nevertheless, it should be noted that a certain percentage of patients considered at baseline time, in a clinical remission status, documented the presence of expressing CMCs (8–50%). A differential expression of the specific transcripts was documented between and within the CMC fractions enriched with CD146- and both CD146/ABCB5-coated beads, confirming the consistency of our approach. Moreover, no molecular expression of our qualitative gene panel was documented in healthy donors’ blood samples subjected to the same enrichment and selection approach, as already described elsewhere.

As described above, we defined and analyzed by Fisher’s exact test the E-CMC and SM-CMC fractions in two distinct disease categories: “clinically remission” patients and “clinically evident disease” patients. We highlighted the following different statistically significant associations comparing molecular expression biomarkers into the two categories for *CD146* 5′-portion (*p* < 0.04), *shCD146* short isoform (*p* < 0.04), and MMP9 (*p* < 0.02) only for the hybrid SM-CMCs subpopulation, while the *lgCD146* isoform was statistically significant in both CMC populations, E-CMCs (*p* < 0.02) and S-M-CMCs (*p* < 0.04). This extended analysis allowed us to “redefine” our previously described data [68], excluding some other genes included in the first molecular expression panel that were significantly associated in both enriched E-CMC and SM-CMC fractions.

So, these findings documented in a more enlarged case series allowed us to develop a highly effective homemade CMC enrichment protocol, selecting CD146 and ABCB5 as melanoma-specific epitopes, even in “clinically remission” patients. Detection of *CD146* 5′-portion, short and long isoforms, and MMPs mostly in enriched hybrid stem–mesenchymal CMCs and even in clinical-remission patients, represented underlying markers of the disease.

## 5. Current Findings: CD146 as an Enrichment and Capture Antigen and Molecular Expression Marker in Magnetically Immune CMC Fractions during Melanoma Follow-Up Time Course

We could also assess a longitudinal monitoring in 13 AJCC staged ≥ pT1b melanoma patients (+6 months ± 48 months), distinguishing them in two disease categories: those who continued or achieved clinical remission (“clinically remission” patients) and those who showed stable or progression disease (“clinically evident disease” patients), as a part of this enlarged project. Demographic and histological characteristics, as well as clinical history of this patient cohort, are reported in Table 3.

Two patients marked with the symbol * (UPN1-AV and UPN2-MU), who presented a longer follow-up period of clinical remission and who subsequently developed progression disease, were evaluated in both the disease categories, as reported in Table 4, part A and Part B. In this regard, despite the case series consisting of 13 patients, we could analyze 15 follow-up time courses.

Molecular analysis was performed in 40 CMC-enriched fractions deriving from the 20 blood samples collected at follow-up, and the 9 transcript-expression frequencies are reported in Table 4A. We analyzed the changes between gene reference status (baseline = positive/negative) and follow-up (positive/negative) by Fisher’s exact test. A shutdown of almost all gene expressions in both subsets was documented when we analyzed the “clinically remission” patient group. In particular, we showed a statistically significant loss of expression of *CD146* (long and short isoforms *p* < 0.004 and *p* < 0.0006, respectively) *N-Cadh* (*p* < 0.04)*, VE-Cadh* (*p* < 0.004), and the two *MMPs (**p* < 0.004). In particular, the short isoform of *CD146*, expressed up to 100% in both the” baseline” subpopulations (E-CMCs and SM-CMCs), resulted in being reduced to zero, showing a strong statistically significant value (*p* < 0.0006).

When analyzing the seven patients (UPN1-AV *, UPN2-MU *, UPN4-VM, UPN7-ZF, UPN 8-GD, UPN9-RETL, and UPN11-DMM) that achieved a clinical remission condition over a period of two years (all undergoing targeted therapy or immune therapy), we documented almost a total negativity expression in all CMC-fractions, if not for some floating “persisting” gene expressions (range 14–42%), as *MMPs*, *N-Cadh*, *VEGF*, and *bFGF,* as reported in Table 4 A. Figure 2 shows an example of a specific melanoma patient’s molecular panel expression (UPN2-MU), previously described and currently updated [70], in which it was possible to observe different distributions of the selected genes between and within CD146- or CD146/ABCB5-coated-CMCs.

The molecular heterogeneity of the two subpopulations at different stages of melanoma disease and during follow-up also was evident for the floating genes. We attempted to interpret these results, at least in part, using the fact that patients undergoing checkpoint-inhibitor therapy or targeted therapy can develop the phenomenon of pseudo-progression.

These findings suggested that CMCs can be efficiently enriched and isolated by either CD146 and/or ABCB5, as melanoma cell-surface markers involved in heterotypic cell adhesion and tissue invasion of melanoma cells. In particular, the contextual CD146 and ABCB5 enrichment identified a hybrid stem–mesenchymal cell population equipped with metastatic abilities such as migration and invasion.

## 6. Soluble CD146 Form in Melanoma Patients

Blot-Chabaud and his group identified, in addition to the membrane-anchored form of CD146, a soluble form (sCD146) [20,35,72,73,74] that is mainly generated by the proteolytic cleavage of the membrane form through metalloproteases [19,35]. As already reported, in healthy people, sCD146 is detected at a concentration of 200 to 400 ng/mL (273 ± 70 ng/mL), and it has been reported to be increased in several diseases, in particular in tumor pathologies [75]. The protein sCD146 is increased in ischemic tissues [35,73,74,75], and appears to enhance neovascularization, vascular regeneration, stabilization, and recovery. Moreover, sCD146 enhances angiogenic properties of endothelium progenitors, playing a major role in trans-endothelial migration. Since a variety of chronic inflammatory diseases are associated with the disruption of the endothelial barrier function, leading to increased permeability, the detection of a high concentration of sCD146 reflects its active role during inflammatory disease as bowel disease [74,75], chronic renal failure [76,77,78], systemic sclerosis [30,35,79,80,81], multiple sclerosis [82], rheumatoid arthritis [83], and other fibrosing autoimmune disorders [84]. We decided to perform a sCD146 dosage by capture enzyme-linked immunoabsorbent assay (ELISA) (CY-QUANT ELISA sCD146, Biocytex, Marseille, France) as follows: 30 melanoma patients’ sera were collected at onset or disease progression; 35 serum samples corresponded to the 13 melanoma patients analyzed at baseline (onset/first observation) and during the follow-up time course analysis. As the control population, we measured sCD146 of seven active metastatic melanoma patients’ sera (IV-AJCC staged) and four patients in a persistent clinical remission status for a long period (up to seven years). All specimens were performed in duplicate, and two distinct experiments were performed in all cases.

Mean/median (SD) sCD146 levels were always concordant when applying the measurement by linear regression or by hyperbolic curve. Unexpectedly, we did not detect values outside the normal range (normal values < 343 ng/mL), with the exception of one serum from a patient with active metastatic melanoma (mean = 562.45 ng/mL; median = 559.25 ng/mL). When analyzing the sCD146 serum concentrations of baseline samples (split into two disease categories: “clinically remission” patients and “clinically evident disease” patients), we documented an increase in the group of patients sharing active evidence of disease with respect to the clinical remission patient sera (Table 2B). Nonetheless, we were able to determine a statistically significant increase in the limit (*p* < 0.051, *t*-test) when we compared the whole group of “baseline patient sera patients” (“clinically remission” patients/“clinically evident disease” patients) vs. the whole “follow-up patients sera” (Table 5 A; Figure 3).

The observed increase in sCD146 in the whole “follow-up group sera”, including both conditions of clinical remission and disease progression, or at least evidence of active disease, prompted us to question whether there was an increase unrelated to the patient’s clinical status. It is known, as already reported, that sCD146 concentration is significantly increased in inflammation, endothelial damage, pathological angiogenesis, autoimmune disorders, and cancer. We subsequently decided to analyze and compare the “follow-up sera”, distinguishing them into distinct clinical remission and evident active disease patient groups. An increase in sCD146 concentration was confirmed in both groups, slight in the clinical remission status, and higher in the group related to evident disease (not statistically significant), as reported in Table 4B and Table 5. A further sCD146 analysis was performed, comparing sera of patients who had not undergone any therapeutic treatment (refusal of therapy or pre-therapy) vs. patients undergoing treatment. In this latter observation, the concentration of sCD146 was substantially increased in both groups, slight in treatment-naïve patients and higher in the treated patient group (not statistically significant), as reported in Table 5. These last two dosage findings are interesting, and could be interpreted while knowing how the immuno-oncological therapies target the immune system and work in an attempt to reactivate it directly in order to eliminate the neoplastic cells, or to inhibit mechanisms of suppression exerted by the tumor. Effectively, targeted therapy and immuno-therapy have revolutionized the treatment of advanced melanoma [85,86].

By using these drugs, it is possible to obtain lasting responses in a large number of patients, and also improve their quality of life for an extended period. During treatment with such immuno-therapy drugs, an increase in the size of the tumor lesions does not necessarily indicate disease progression and the consequent mandatory passage to another line of therapy. Known as “pseudo-progression”, this process can occur earlier or later in the course of treatment, when the T cells infiltrate the neoplastic sites, and can be followed, for a second time, by neoplastic regression. It has been hypothesized that in the case of early pseudo-progressions, the cells of the immune system may initially increase the volume of the lesions, or the activation of the immune system may take longer: the tumor volume may then initially increase and then regress when the antitumor immune response becomes effective. On the other hand, late pseudo-progressions can be explained by hypothesizing that the balance between the immune system and the tumor are dynamic and long-term processes, which can result in an undulating clinical effect of tumor regression and growth [87,88,89,90]. In this sense, sCD146 does not seem to have a role as a specific diagnostic marker, but rather as a marker-index of inflammation and/or tissue damage, and an appealing marker for disease monitoring. The increased values of sCD146 during therapeutic treatment could be explained by the pseudo-progression derived from it. This process could also explain the fluctuating molecular positive expression of some genes (*N-Cadh*, *VEGF*, *bFGF,* and *MMPs*) highlighted during sequential monitoring in the cellular fractions under analysis, with some of these effectively not associated with any statistically significant value.

Two other considerations emerged from our sCD146 dosages. Firstly, it will be mandatory to confirm these data by analyzing in larger case series, including the condition of active disease, the evaluation of clinical remission status, and the therapy time course monitoring. The second consideration led us to propose to lower the limit of the “normal” range corresponding to the healthy population, now fixed as 273 ± 70 ng/mL, at least in the context of malignant melanoma, and establishing a lower cut-off at around 343 ng/mL.

## 7. Discussion

In recent years, molecular oncology has been focused on pathways of cancer metastasization and molecular mechanisms of tumor-cell spreading. The results showed that invasion also may occur in early tumor development, as well as in supposed dormant tumors [91,92,93,94]. Tumor-induced angiogenesis occurs together with the transition to invasion, providing a vascular network suitable for dissemination. These phenomena can precede growth of the primary tumor outgrowth by several years. The main mechanism of tumor dissemination is the ability of cell to perform the epithelial-to-mesenchymal transition (EMT), which is mandatory for tumor invasion and metastasis [95,96]. Epithelial cells play a structural and functional role achieved by cell-to-cell and cell-to-extracellular-matrix junction adhesions. This process is coordinated by cadherins and integrins connected by keratin intermediate filaments (KIFs), resulting in a continuous cytoskeleton that connects the keratinocytes of the skin. This structure gives flexibility and structural rigidity to the whole tissue. On the other hand, mesenchymal cells appear structurally relaxed, and are prone to mobility. EMT implicates the downregulation or loss of some epithelial cell markers, such as epithelial cadherin or keratins, and their replacement with mesenchymal cell markers such as vimentin or neuronal cadherin [97,98,99,100,101,102,103,104].

Once they have reached a targeted organ, mesenchymal-like tumor cells may also reverse to epithelial-like tumor cells, by the mesenchymal-to-epithelial transition (MET), regaining their epithelial phenotype and losing their migratory phenotype [101,105]. So, EMT and MET together seem to be necessary for the establishment and development of a clinically relevant metastasis [101,102,103,104,105].

Recent studies showed that maintenance in a hybrid epithelial/mesenchymal (E/M) state is sufficient for maintaining stem cell properties with a wide phenotypic plasticity, spanning from an epithelial to a total mesenchymal stage. This passing through a number of intermediate hybrid stages allows these cells to move collectively and to efficiently reach the bloodstream, giving rise to clusters of circulating tumor cells (CTCs) and forming metastases [102,103,104,105].

The specific detection of CTCs and of circulating tumor microemboli has acquired high importance, due mainly to technological improvements leading to the identification and isolation of these “rare” cellular elements. The liquid biopsy today represents a new tool in the clinical management of oncological patients. This method consists of isolating rare CTCs in peripheral blood.

Several studies have documented circulating melanoma cells (CMCs), even in early stages (AJCC classification) [106,107,108,109,110].

Characterizing the epithelial vs. mesenchymal phenotypes of CMCs may be helpful to identify the aggressive CTC subpopulations and establish a valid therapy [111,112,113,114,115]. We developed a highly effective CMC enrichment protocol [68,70] by selecting a reference gene panel. We enriched and analyzed CMCs from melanoma patients (AJCC staged ≥ pT1b) using CD146 and ABCB5 as melanoma-specific epitopes [68,70]. Tumorigenic heterogeneity within the melanoma vertical growth phase (VGP) is due to a subpopulation of human melanoma cells expressing the multidrug resistance transporter, ABCB5 [71,110,116]. The ABCB5 trans-membrane transporter is strongly associated with melanoma genesis, stem cell maintenance, metastasis, and chemoresistance [71,110]. We selected CD146 for its high surface expression, up to 80% [1,2,3,4,5,6,8,9,10,11,12,13,58,59,68,70], and because it is considered as an EMT inducer [117,118,119,120]. In addition, it functions as key oncogene in driving melanoma progression and metastasis, and represents a mesenchymal marker [117,118,119,120,121,122,123,124,125]. Here, we present the results of our studies, in which it was possible to expand the number of cases to be analyzed, confirming that CD146 and ABCB5 were suitable and effective cell-surface targets using our analysis system. The different expression of the specific transcripts, documented between and within the two CMC fractions, confirmed the validity of this approach, outlining different specific patient expression profiles. We obtained a sensitivity of 90% (27 out of 30) when analyzing the “baseline samples-CMCs”, considering that 40% (12 patients out of 30) were in clinical remission, and despite showing variable molecular expressions (8–50%).

Our method that conjugated CD146 and ABCB5 antibodies identified one CMC-fraction, the “hybrid stem–mesenchymal”, which well-defined the actively metastasizing cells that were characteristic of the active disease. Our gene panel documented that only *CD146* 5′-*portion, short* and *long* isoforms, and *MMP9* represented statistically significant disease markers, even in patients defined as in clinical remission. The endothelial fraction showed a statistically significant association only for the *lgCD146* isoform. Moreover, these data allowed us to further restrict the number of genes to be “analyzed” at the baseline time. MMP9 molecular expression represents a “warning biomarker of disease”. MMP-9 plays a prominent role in angiogenesis by cooperating with VEGF [126,127,128], and it is produced by metastatic tumor cells and metastasis-infiltrating neutrophils and/or macrophages. The mechanism involves the breakdown of the vessel basement membrane and perivascular matrix, thereby generating extracellular matrix (ECM) fragments [128,129,130,131]. At the same time, MMP-9 retrieves ECM-bound VEGF in a soluble form, which mediates endothelial cell proliferation. MMP-9 is also important in the angiogenic switch that causes either the growth or the metastatic dissemination of the primary tumor [130,131]. Moreover, in tumor tissue, MMP-9 triggers the formation of new lymphatic vessels, providing additional routes for cancer metastasis [131].

Our data on the sequential time course of 13 melanoma patients (effectively, as reported, 15 time-course analyses), showed that the two distinct CMC subpopulations, “endothelial” and “hybrid stem–mesenchymal” fractions, expressed *CD146* isoforms *long, short,* or *5′-portion* at first observation (from 37.5% to 100%) and at disease progression (100%).

We documented a statistically significant repression of *CD146* (*long* and *short* isoforms), *N-Cadh, VE-Cadh,* and *MMPs* transcripts when patients achieved clinical remission. Particularly, the *shCD146* isoform, expressed up to 100% in both ”baseline” subpopulations (E-CMCs and SM-CMCs), was totally ablated (*p* < 0.0006). Importantly, the shCD146 antigen, expressed in the apical pole of the cell, contributed to tumoral angiogenesis, and its regulation was more consistent with respect to other proangiogenic factors such as *VEGF* and *bFGF,* displaying a fluctuating expression during therapeutic treatment.

We analyzed the circulating soluble form of the CD146 protein by using a specific ELISA Kit, the sCD146. We demonstrated at first that it constituted an active factor, playing a major role in angiogenesis. This effect seemed to be mediated by its binding to the p80 isoform of angiomotin [1,132]. This protein was detectable on the vasculature of ischemic tissues, but also on many tumor cells [133,134,135], suggesting that CD146-positive tumors could secrete soluble CD146 involved in their growth and vascularization. The sCD146 secreted by CD146-positive tumors did not only display effects on tumor angiogenesis, but also on tumor growth and survival. Interestingly, as already reported, the sCD146 concentrations increased in several diseases, in particular in tumors [73,75,133,134,135,136]. We confirmed this trend, detecting an increase in sCD146 in the “follow-up group sera”, in both conditions of clinical remission and disease progression. To explain this result, we should take into consideration the immunotherapy treatment of the melanoma.

The observed sCD146 increase found in patient’s sera undergoing therapeutic treatment could be explained as a “phlogistic/inflammatory response” related to a pseudo-progression widely observed during immunotherapy. It could represent a marker for disease monitoring [85,88,89,137,138,139,140,141], suggesting an important role of this form of the protein as an inflammation marker.

These considerations require a wider confirmation and validation with a longitudinal sequential analysis over a longer time.

According to our findings and those in the literature, CD146 can be considered as a membrane antigen suitable for identification and enrichment in melanoma liquid biopsies, as a highly effective molecular “warning marker for melanoma minimal residual disease monitoring“ [142], and finally as a soluble protein index of inflammation and putative response to therapeutic treatments. Notably, the targeting of both sCD146 and CD146 by monoclonal antibodies able to neutralize its effects could constitute an innovative antitumoral strategy.

Our molecular qualitative reference gene panel now has been validated in a larger case series, achieving a stronger statistical significance for some genes such as *CD146 short* and *long* isoforms, *N-Cadh*, *VE-Cadh*, *MMP2,* and *MMP9*. Effectively, it is suitable to identify those genes that could provide great potential and biological information to better define melanoma high-risk and low-risk patients, and the most important role seems to be played by CD146. Actually, we are assessing a quantitative real-time polymerase chain reaction (qRT-PCR) to define mRNA measurement detection thresholds, particularly for those reference genes, found to have a statistically significant association with disease progression.

Taken together, CD146 molecular expression analysis at onset or at disease recurrence, the sequential monitoring of the statistically significant transcripts, and the detection of the soluble form could help to follow the melanoma remission or progression, even in apparent disease-free status. A representative scheme of our current method for detection, isolation, and enrichments of actively metastasizing CMCs is given in Figure 4.

## Figures and Tables

**Figure 1 ijms-22-12416-f001:**
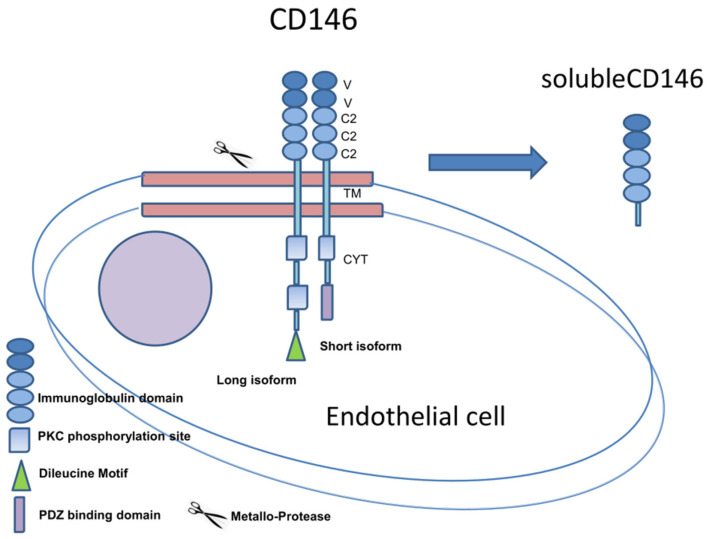
Schematic Representation of CD146 Antigen as membrane protein and soluble isoform.

**Figure 2 ijms-22-12416-f002:**
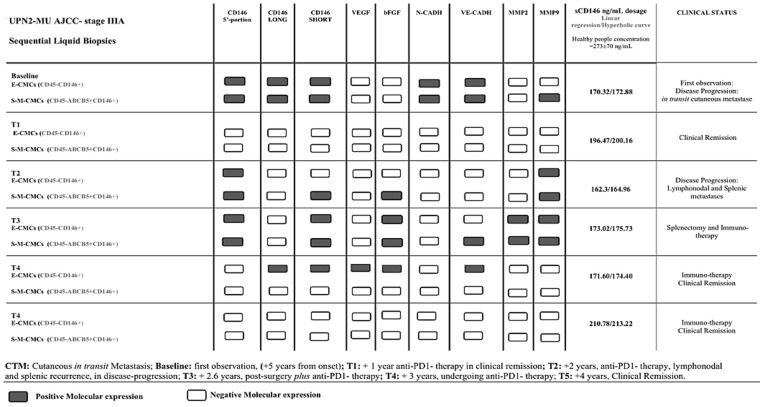
Specific patient’s molecular panel expression.

**Figure 3 ijms-22-12416-f003:**
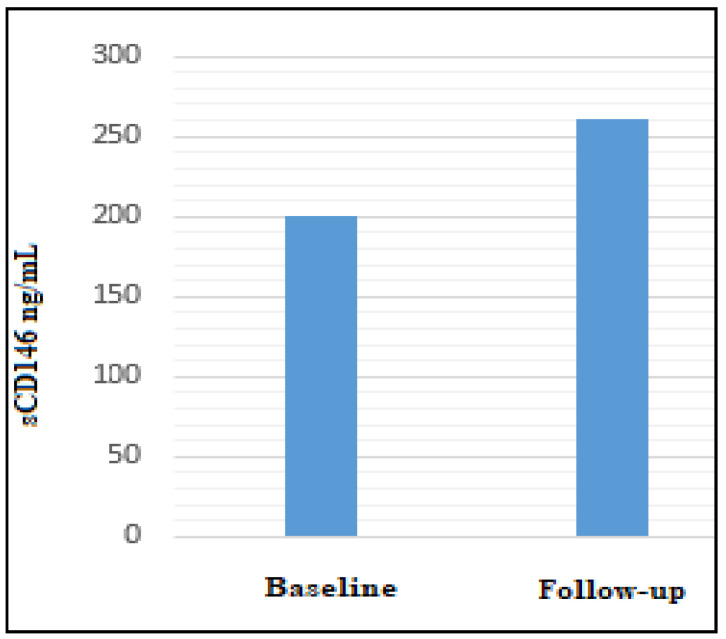
Bar graph showing the distribution of sCD146 concentrations in the two patient sera groups: baseline vs. follow-up (*p* < 0.051).

**Figure 4 ijms-22-12416-f004:**
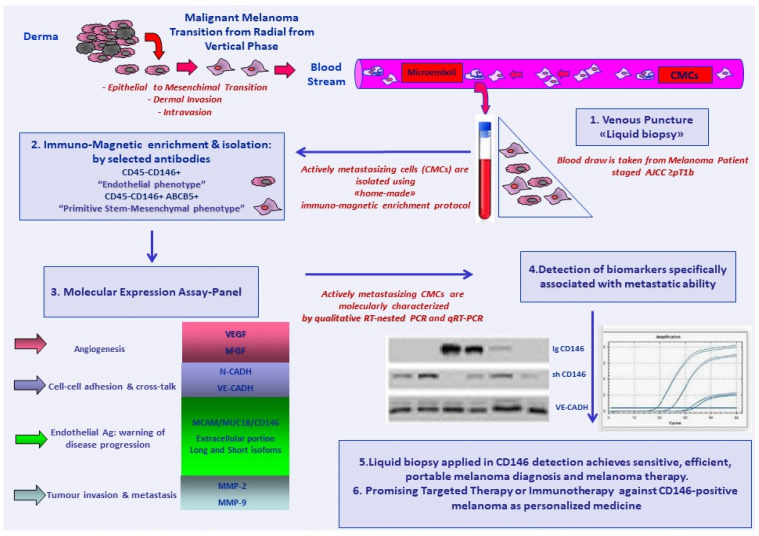
Schematic diagram of our current method for detection, isolation and enrichments of actively metastasizing CMCs.

**Table 1 ijms-22-12416-t001:** Patients’ (pts) demographic and clinical characteristics.

**Sex**	**N°**	**%**
Female	13	43.33
Male	17	56.66
Age (Years)	44 (mean)	23–84 (range)
**Primary Tumour Site**	**N°**	**%**
Head and Neck *	2	6.67
Trunk	17	56.67
Extremity	7	23.33
Unknown	4	13.33
**AJCC** **** Stage**	**N°**	**%**
>IB	5	16.67
II: IIA (3); IIB (2);	5	16.67
III ***: IIIA (2); IIIB (3); IIIC (2);	7	23.33
IV	13	43.33
**Time from Diagnosis**	**Years**	**Pts/AJCC Stage**
Onset	0–11	18: IB (5); IIA (3); IIB (2); IIIA (1); IIIB (1); IIIC (2); IV (4)
First Observation–Baseline		12: IIIA (1); IIIB (2); IV (9)
**Clinical Status**		**%**
Clinically Disease-Free	12	40
Clinically Evident Disease	18	60

* Mucosal melanoma of the nasal cavity. ** The AJCC (American Joint of Cancer Committee) staging was evaluated at the time of the blood draw after diagnosis of primary melanomas or diagnosis of first distant metastases in the case of occult melanomas. *** Three out of five pts showed cutaneous in transit metastasis without nearby lymph node involvement (N1c).

**Table 2 ijms-22-12416-t002:** (A) Molecular expression (percentage) of proangiogenic markers, cell–cell adhesion factors, and matrix metalloproteinases in the two enriched CMC subpopulations from melanoma patients collected at disease onset or first clinical observation (baseline) and soluble CD146 (sCD146) serum dosages are reported here in two classes of patients distinguished by their clinical status.

Clinically Remission Expression Panel Patients Baseline Samples (#12)	CD146 5′-Portion	CD146 Long Isoform	CD146 Short Isoform	VE-Cadh	N-Cadh	VEGF	b-FGF	MMP-2	MMP-9
**Endothelial CMCs (E-CMCs)** **(CD45-MCAM/CD16 +)**	**16.6%** 2/12	**33.3%** 4/12	**33.3%** 4/12	**25%** 3/12	**33.3%** 4/12	**8.3%** 1/12	**33.3%** 4/12	**41.6%** 5/12	**41.6%** 5/12
**Stem-Mesenchimal CMCs (S-M-CMCs)** **(CD45-MCAM + ABCB5 +)**	**8.3%** 1/12	**25%** 3/12	**16.6%** 2/12	**33.3%** 4/12	**25.0%** 3/12	**16.6%** 2/12	**16.6%** 2/12	**50.0%** 6/12	**33.3%** 4/12
**(A)** **Frequency of expression (%) for selected genes** (number of positive CMCs subpopulations for selected genes/total number of patients series)									
**Clinically Evident Disease Patients Expression Panel** **Baseline Samples (#18)**									
**Endothelial CMCs (E-CMCs)** **(CD45-MCAM/CD16 +)**	**44.4%** 8/18	**72.2%** 13/18	**61.1%** 11/18	**55.5%** 10/18	**33.3%** 6/18	**16.6%** 3/18	**11.1%** 2/18	**66.6%** 12/18	**55.5%** 12/18
**Stem-Mesenchimal CMCs (S-M-CMCs)** **(CD45-MCAM + ABCB5 +)**	**50.0%** 9/18	**77.7%** 14/18	**72.2%** 13/18	**55.5%** 10/18	**27.7%** 5/18	**5.5%** 1/18	**22.2%** 4/18	**66.6%** 12/18	**72.2%** 13/18

**Table 3 ijms-22-12416-t003:** Patients’ demographic and histological characteristics, and clinical history.

UPN	Sex	Age at Baseline	Primary Tumor Site	Histology	Breslow Grade (mm)	AJCC Status at Baseline	Incurrence of Progression from Diagnosis	Therapy after Diagnosis or Lymphnodal/Cutaneous in Transit/Metastases Incurrence	Follow-Up and Clinical Status
**UPN1-** **AV**	f	80	Unknown	/	/	IV	+1 year	Checkpoint inhibitors (anti-PD1-PD1L)	At +2 years: disease progression Actually in therapy
**UPN2-** **MU**	m	47	Trunk	SSM	1.8	IIIA	+5 years	Targeted therapy (anti-BRAF and anti-MEK)	At +4 years: disease progression Actually clinical remission
**UPN3-** **FM**	m	40	Trunk	SSM	1.25	IIB	+3 years	Pretargeted therapy (anti-BRAF and anti-MEK)	At +1 year: stable disease Actually in therapy
**UPN4-** **VM**	m	60	Trunk	NM	4.5	IIB	+1 year	Checkpoint inhibitors (anti-PD1-PD1L)	At +2 years: continuous clinical remission
**UPN5-** **CAD**	f	82	Acral	NM	2.4	IV	+13 years	Checkpoint inhibitors(anti-PD1-PD1L)	At +1 year: disease progression and death
**UPN6-** **PN**	m	64	Trunk	NM	4.0	IV	+1 year	Targeted therapy (anti-BRAF and anti-MEK)	At +1 year: disease progression and death
**UPN7-** **ZF**	m	42	Head	NM	2.2	IB	/	Checkpoint inhibitors (anti-PD1-PD1L)	At +2 years: continuous clinical remission
**UPN8-** **GD**	m	35	Acral	NM	2.2	IIA	+7 years	Targeted therapy (anti-BRAF and anti-MEK)	At +1 year: continuous clinical remission
**UPN9-** **RETL**	f	53	Acral	ALM	/	IIIC	/	Targeted therapy (anti-BRAF and anti-MEK)	At +2 years: clinical remission Actually stopped therapy
**UPN10-** **PME**	f	75	Nasal cavity	Mucous MM	/	IIB	/	Checkpoint inhibitors (anti-PD1-PD1L)	At +6 months: disease progression Actually in therapy
**UPN11-** **DMM**	f	34	Trunk	NM	1.5	IIIA	+2 years	IFN	At +6 years: continuous clinical remission Actually stop-therapy
**UPN12-** **SD**	f	41	Trunk	NM	7	IIB	+2 years	Checkpoint inhibitors (anti-PD1-PD1L)	At +4 years: disease progression Actually in therapy
**UPN13-** **BC**	m	86	Trunk	Trunk	5	IIB	+4 years	Refusal of any therapy	At +1 year: stable disease

**Table 4 ijms-22-12416-t004:** Molecular expression (percentage) of proangiogenic markers, cell–cell adhesion factors, and matrix metallo-proteinases in the two enriched CMC subpopulations from the melanoma patients in clinical remission status, (Part A) and melanoma patients in stable or disease progression clinical status, (Part B). Because two patients* (UPN1-AV and UPN2-MU) with a longer follow-up after a period of clinical remission developed disease progression, the study included and analyzed both the two follow-up conditions. Consequently, despite the case series consisting of 13 patients, it is reported here the sequential molecular expression of 15 follow-up blood samples.

**(A)**									
*** Clinically Remission Patients**	**CD146** **5′-Portion**	**CD146 Long Isoform**	**CD146 Short Isoform**	**VEGF**	**bFGF**	**N-Cadh**	**VE-Cadh**	**MMP-2**	**MMP-9**
**Baseline Expression Panel (#7)**									
**Endothelial CMCs** **(E-CMCs)** **(CD45-MCAM +)**	**42.8%** 3/7	**85.7%** 6/7	**100%** 7/7	**57.1%** 4/7	**28.6%** 2/7	**71.4%** 5/7	**85.7%** 6/7	**85.7%** 6/7	**71.4%** 5/7
**Stem-Mesenchimal CMCs** **(S-M-CMCs)** **(CD45-MCAM + ABCB5+)**	**42.8%** 3/7	**85.7%** 6/7	**100%** 7/7	**28.6%** 2/7	**57.1%** 4/7	**57.1%** 4/7	**71.4%** 5/7	**85.7%** 6/7	**71.4%** 5/7
**Follow-Up Expression Panel (#7)**									
**Endothelial CMCs** **(E-CMCs)** **(CD45-MCAM +)**	**0%** 0/7	**0%** 0/7	**0%** 0/7	**14.3%** 1/7	**14.3%** 1/7	**42.8%** 3/7	**0%** 0/7	**28.6%** 2/7	**28.6%** 2/7
**Stem-Mesenchimal CMCs** **(S-M-CMCs)** **(CD45-MCAM + ABCB5+)**	**0%** 0/7	**0%** 0/7	**14.3%** 1/7	**0%** 0/7	**14.3%** 1/7	**0%** 0/7	**0%** 0/7	**0%** 0/7	**0%** 0/7
**(B)**									
**Baseline Expression Panel (#8)**									
**Endothelial CMCs** **(E-CMCs)** **(CD45-MCAM+)**	**37.5%** 3/8	**62.5%** 5/8	**50.0%** 4/8	**12.5%** 1/8	**25.0%** 2/8	**25.0%** 2/8	**50.0%** 4/8	**62.5%** 5/8	**50.0%** 4/8
**Stem-Mesenchimal CMCs** **(S-M-CMCs)** **(CD45-MCAM + ABCB5+)**	**50.0%** 4/8	**75.00%** 6/8	**62.5%** 5/8	**25.0%** 2/8	**50.0%** 4/8	**37.5%** 3/8	**62.5%** 5/8	**62.5%** 5/8	**87.5%** 7/8
**Follow-Up Expression Panel (#8)**									
**Endothelial CMCs (CD45-MCAM + E-CMCs)**	**37.5%** 3/8	**62.5%** 5/8	**100%** 8/8	**25.0%** 2/8	**25.0%** 2/8	**50.0%** 4/8	**62.5%** 5/8	**62.5%** 5/8	**87.5%** 7/8
**Stem-Mesenchimal CMCs** **(S-M-CMCs)** **(CD45-MCAM + ABCB5+)**	**50.0%** 4/8	**87.5%** 7/8	**62.5%** 5/8	**37.5%** 3/8	**12.5%** 1/8	**25.0%** 2/8	**62.5%** 5/8	**62.5%** 5/8	**100%** 8/8

**Table 5 ijms-22-12416-t005:** Soluble CD146 (sCD146) serum dosages and graphics.

Comparison Between Clinical Serum Classes	sCD146 ng/mL Dosage Mean/Median Healthy People Concentration = 273 ± 70 ng/mL
**(A)** **Melanoma Baseline (Onset/First Observation) Sera** (#30)	199.55/186.78 Linear regression 201.01/191.40 Hyperbolic curve
**Melanoma Follow Up Sera** (#21)	258,28/230.60 Linear regression 260.26/232.85 Hyperbolic curve
**(B)** **Clinically Remission Patient Sera** (#13)	212.651/197.48 Linear regression 215.72/203.255 Hyperbolic curve
**Clinically Evident Disease Patient Sera** (#22)	262.13/259.16 Linear regression 263.88/261.22 Hyperbolic curve
**(C)** **Treatment-Naïve Patient Sera** (#12)	217.58/218.68 Linear regression 198.5/206.35 Hyperbolic curve
**Treated-Patient Sera** (#22) (Checkpoint Inhibitor Therapy–Targeted Therapy–Other Therapies)	253.85/235.33 Linear regression 255.87/237.82 Hyperbolic curve

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
