# Peer review of "MCAM/MUC18/CD146 as a Multifaceted Warning Marker of Melanoma Progression in Liquid Biopsy"

_ijms, 2021, doi:10.3390/ijms222212416_

Round 1

Reviewer 1 Report

In the present manuscript, the authors review the role of the molecular expression of MCAM/MUC18/CD146 in the onset and recurrence of melanoma disease in different cohorts of patients. The authors also discuss the suitability of MCAM/MUC18/CD146 to be considered as 1/a membrane-antigen suitable for identification and enrichment in melanoma liquid-biopsy, 2/a high effective molecular “warning” marker for minimal residual disease monitoring, and 3/a soluble protein index of inflammation and putative response to therapeutic treatments

It is a very interesting review that can help to improve the monitoring of the progress of melanoma. As such, it is suitable for publication in International Journal of Molecular Sciences in its present form, however, prior to acceptance, the manuscript needs some changes in order to improve the quality of the manuscript.

Major comments:

In my opinion, it would be interesting to add a simple diagram of the structure of CD146 due to the complex nature of the protein.

Table 1 is well prepared and is perfectly understood except for the last two rows in which the data are not well aligned and lead to confusion. I think they should be aligned correctly so as not to be confusing.

Table 2 is very confusing in my opinion. I do not understand why it is separated into A and B since part B is placed as part of A. Also, if the authors refer to the protein as CD146 in the manuscript, I do not understand why they write MCAM/MUC18/CD146 in the tables. They should keep the criteria. Regarding the serum levels of CD146, it would be preferable to provide one of the data either by linear regression or by hyperbolic curve. Both approaches are acceptable and give very similar results, so giving both data only adds confusion to the table. Authors should rebuild the table 2 to make it easier to understand.

Table 4 is also very confusing and presents the same problems as table 2. Authors should rebuild the table 2 to make it easier to understand.

In figure 1 the legend of “slightly positive molecular expression” is not well distinguished from the negative molecular expression. The authors should put another legend that does not cause confusion.

Figure 3 should be figure 2 due to its position in the text of the manuscript. Error bars and statistical significance are missing from the bar chart. In my opinion the graphs of the linear regression and the hyperbolic curve do not add anything to the manuscript. I think these graphics could be removed.

Minor comments:

-The manuscript contains some typographical errors e.g. on page 2 line 95.

-The manuscript contains one grammatical error on page 4 line 203.

Author Response

To the editorial Board of

International Journal of Molecular Science

 Thank you for offering us a great opportunity, to upload our manuscript to your journal especially because thanks to your recommendations, we could see our work published in a well quoted scientific journal.  We send to you the responses as requested by your “Comments and Suggestions for Authors
”.           

Reviewer 1

In the present manuscript, the authors review the role of the molecular expression of MCAM/MUC18/CD146 in the onset and recurrence of melanoma disease in different cohorts of patients. The authors also discuss the suitability of MCAM/MUC18/CD146 to be considered as 1/a membrane-antigen suitable for identification and enrichment in melanoma liquid-biopsy, 2/a high effective molecular “warning” marker for minimal residual disease monitoring, and 3/a soluble protein index of inflammation and putative response to therapeutic treatments

It is a very interesting review that can help to improve the monitoring of the progress of melanoma. As such, it is suitable for publication in International Journal of Molecular Sciences in its present form, however, prior to acceptance, the manuscript needs some changes in order to improve the quality of the manuscript.

Major comments:

  • In my opinion, it would be interesting to add a simple diagram of the structure of CD146 due to the complex nature of the protein.

We included a schematic simplified image of CD146 antigen, protein belonging to the Immunoglobulin-superfamily as requested and reported as Figure 1.

  • Table 1 is well prepared and is perfectly understood except for the last two rows in which the data are not well aligned and lead to confusion. I think they should be aligned correctly so as not to be confusing. Table 2 is very confusing in my opinion. I do not understand why it is separated into A and B since part B is placed as part of A. Also, if the authors refer to the protein as CD146 in the manuscript, I do not understand why they write MCAM/MUC18/CD146 in the tables. They should keep the criteria. Table 4 is also very confusing and presents the same problems as table 2. Authors should rebuild the table 2 to make it easier to understand.

Effectively, we had sent figures and tables as separate files in word and jpeg format, hoping that they could be processed and inserted in the main-text, by the editorial platform. But they are obviously totally not formatted as intended. In fact we will send figures and tables as separate files in word and j-peg format, hoping that they could be processed and inserted in the main-text, by the editorial platform. But they turned out to be confused and completely "disrupted", without the lines and paragraphs, once inserted and transformed into PDF format , obviously totally not formatted as intended. We have made arrangements with Slavomir Nikodijevic Assistant Editor- MDPI and will provide us with IT support.

  • Regarding the serum levels of CD146, it would be preferable to provide one of the data either by linear regression or by hyperbolic curve. Both approaches are acceptable and give very similar results, so giving both data only adds confusion to the table. Authors should rebuild the table 2 to make it easier to understand.

Serum levels of CD146: the choice to show both the obtained measurements (hyperbolic and linear regression) is due to the fact that the assay performed showed a "naturally" hyperbolic trend. However, the kit provided by Prof. Blot-Chabaud suggests a transformation of the assays to linear regression. Although the sCD146 ELISA kit is commercially available, the dosage to date is neither routine nor standardized in a ubiquitous way (each laboratory today uses its own internal controls), so we prefer to show both the dosages, since, as you also pointed out, “both approaches give very similar confident results”. We believe that re-presenting in the original correct version the tables 2,4 and 5, the data can be more understandable and appreciable.

  • In figure 1 the legend of “slightly positive molecular expression” is not well distinguished from the negative molecular expression. The authors should put another legend that does not cause confusion.

We modified the Figure 1, actually renamed Figure 2 avoiding to show the slightly molecular expression "unnecessary and confusing” as you remarked.

  • Figure 3 should be figure 2 due to its position in the text of the manuscript. Error bars and statistical significance are missing from the bar chart. In my opinion the graphs of the linear regression and the hyperbolic curve do not add anything to the manuscript. I think these graphics could be removed.

We replaced and renamed all figures, since we also included a new figure (Figure 1, the CD146 image) as you requested.

  • Minor comments:

-The manuscript contains some typographical errors e.g. on page 2 line 95.

-The manuscript contains one grammatical error on page 4 line 203.

We have introduced corrections and changes to the text with red characters.

Reviewer 2 Report

This study presents importance of MCAM/MUC18/CD146 in melanoma patient followup using concepts of liquid biopsy. Essentially, authors re-establish and strengthen the importance of this biomarker in minimal disease residual monitoring, and in inflammation and response to therapy.   

This is an extension of their previous study on a larger pool of patients. On a small numbers of patients, authors have presented a time-course analysis, which is important for clinical usage. 

The manuscript could be improved by reducing the excessive introduction. I understand the manuscript is written as a review article, but this confounds the reader on the additional results presented in this manuscript. Authors may also want to include p-values in the tables itself. 

Perhaps it is out of scope, but it would be interesting to see if this concept could be used for both targeted therapy and immunotherapy. Also, authors may want to discuss about tumour mutation burden and if this is associated with the CD146 expression.

Author Response

To the editorial Board of

International Journal of Molecular Science

 Thank you for offering us a great opportunity, to upload our manuscript to your journal especially because thanks to your recommendations, we could see our work published in a well quoted scientific journal.  We send to you the responses as requested by your “Comments and Suggestions for Authors
”.           

Reviewer 2

This study presents importance of MCAM/MUC18/CD146 in melanoma patient follow up using concepts of liquid biopsy. Essentially, authors re-establish and strengthen the importance of this biomarker in minimal disease residual monitoring, and in inflammation and response to therapy.   

This is an extension of their previous study on a larger pool of patients. On a small numbers of patients, authors have presented a time-course analysis, which is important for clinical usage. 

  • The manuscript could be improved by reducing the excessive introduction. I understand the manuscript is written as a review article, but this confounds the reader on the additional results presented in this manuscript. Authors may also want to include p-values in the tables itself

We reduced Introduction as requested. We actually decided to write a review on the important and multifaceted role of the CD146 antigen, reporting what we have achieved in the literature and our experience to date.  We emphasize the suitability of MCAM/MUC18/CD146 as “1/a membrane-antigen suitable for identification and enrichment in melanoma liquid-biopsy, 2/a high effective molecular “warning” marker for minimal residual disease monitoring, 3/a soluble protein index of inflammation and putative response to therapeutic treatments. “

Besides having reviewed the topic, we are aware having also added new data from expanded analyzed series that we believe can reinforce the description.

The Fisher exact-text documented statistical significant p-values of some of these bio-markers, when comparing the enriched two subpopulations  (E- CMC and SM CMC  fractions)  in two distinct disease categories: clinically disease free patients” and “clinically evident patients” at baseline or during follow-up. As reported we analyzed the changes between gene-reference- status (baseline = positive/negative) and follow-up (positive/negative) by Fisher’s-exact test, obtaining statistically significant data. The data presented in our tables are shown to report and highlight the molecular expression (percentages) of the our bio-markers in the two subpopulations of enriched  circulating cells, at different times and stages of disease: at baseline and during follow up in condition of clinical remission or in evident disease.Entering the p-value in our opinion would make it more complex to set up our data and would “distort” what was intended to be presented in the two tables. However, we highlighted with asterisks those markers that were statistically significant in tables 2 and 4 by referring to the main text for the p-value.

  • Perhaps it is out of scope, but it would be interesting to see if this concept could be used for both targeted therapy and immunotherapy. Also, authors may want to discuss about tumour mutation burden and if this is associated with the CD146 expression.

It could be an interesting and exciting suggestion initiate a new line of translational research by analyzing, at the same time the mutational burden state of patients and comparing it to molecular expression of this antigen and the analysis of its soluble counterpart, but it is out of our current experience.
